# Accelerating AI Performance using Anderson Extrapolation on GPUs

**Saleem Abdul Fattah Ahmed Al Dajani**\*,  **David E. Keyes**

Applied Physics Program, Physical Sciences and Engineering Division
Extreme Computing Research Center, Computer, Electrical and Mathematical Sciences and Engineering Division
King Abdullah University of Science and Technology (KAUST)
Thuwal, Makkah Province, Kingdom of Saudi Arabia (KSA) 23955-6900
`saleem.aldajani@kaust.edu.sa`, `david.keyes@kaust.edu.sa`

## Abstract

We present a novel approach for accelerating AI performance by leveraging Anderson extrapolation, a vector-to-vector mapping technique based on a window of historical iterations. By identifying the crossover point (Fig. 1) where a mixing penalty is incurred, the method focuses on reducing iterations to convergence, with fewer more compute-intensive but generally cacheable iterations, balancing speed and memory usage with accuracy and algorithmic stability, respectively. We demonstrate significant improvements in both training and inference, motivated by scalability and efficiency extensions to the realm of high-performance computing (HPC).

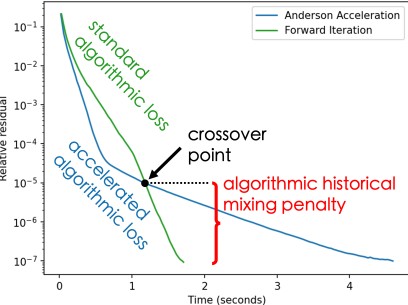

Figure 1: Crossover and mixing penalty plotted against time. Relative residual is $\frac{\|f(z^k,x)-z^k\|_2}{\|f(z^k,x)\|_2+\lambda}$ [35].

## 1   Introduction

Anderson extrapolation [1, 2, 16, 27, 30, 34] has recently been applied to deep equilibrium models (DEQs) [7–10, 17, 24]. Kolter et al. [35] found the gains not substantial due to early termination with a loose convergence tolerance. They focused on Anderson extrapolation during training. Here, we show significant acceleration of AI performance with Anderson on GPUs for both the forward pass (running inferences faster) and training (generating models faster). We demonstrate acceleration of the forward pass with standard Anderson as a baseline for future work with stochastic variants [31] and accelerating the backward pass with Jacobian-free methods like Jacobian-Free Backpropagation (JFB) and Neumann series gradient approximations [16].

As AI demand grows, as shown in Fig. 2 [3, 15, 25, 28], high-performance computing (HPC) is becoming critical due to economic pressures from the growth of data and AI infrastructure [29]. Low-memory acceleration techniques, like Anderson extrapolation, will be key to increasing HPC-based AI computational efficiency. This study investigates matrix-free Anderson extrapolation on GPUs, emphasizing gains from advanced computing architectures compared to CPUs. Our

---

\*Code can be found at http://tinyurl.com/DeepAndersoNN

38th Second Workshop on Machine Learning with New Compute Paradigms at NeurIPS 2024(MLNCP 2024).

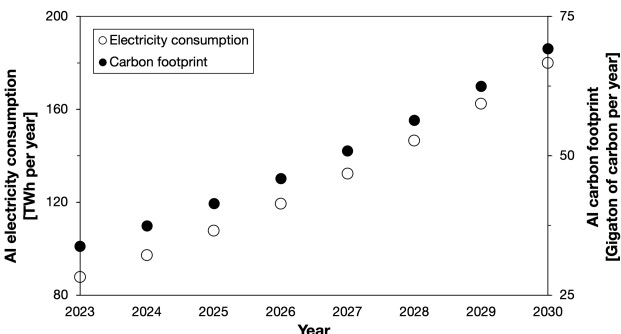

Figure 2: AI carbon footprint projected to consume >2% of global electricity demand [3, 15, 25, 28], amounting to >10% of global electricity demand for data centers and infrastructure.

goal is to maximize computational efficiency while reducing iterations to convergence by reusing previous iterations to avoid unnecessary gradient calculations, gaining partial benefits expected from second-order methods (e.g., [33]) without manipulating Hessian matrices.

The environmental impact of AI is rapidly growing [3, 15, 25, 28]. By 2030, AI is projected to account for 2% of global electricity consumption. We aim to reduce this impact by up to 90%, saving 160 terawatt-hours per year by 2030. The carbon footprint of AI exceeds the 500-megaton annual benchmark set by initiatives like Bill Gates' Breakthrough Energy [14]. Efficiency-enhancing technologies like GPU and Anderson acceleration can reduce AI's carbon emissions by 60 gigatons per year by 2030, as shown in Fig. 2.

## 1.1 Leveraging extrapolation for AI and HPC advances

Anderson extrapolation, a windowing technique for accelerating nonlinear fixed point iterations diagrammed in Figs. 3 and 4, is widely applied in fields like density functional theory, kinetic theory, and climate spin-up. It is well-suited for distributed memory parallelization and GPU implementation. It is a staple of major open-source large-scale solver libraries, including PETSc [11, 12], SUNDIALS [23], Trilinos [19–22], and deal.II [4–6, 13]. It can be applied to machine learning training, smoothing out standard forward iterations and achieving superior accuracy in training and testing error. Benchmarking results on CIFAR10 show expected robustness benefits and allow characterization of the temporal advantages or disadvantages from the higher cost per iteration, where a small residual minimization step is applied at each new function evaluation.

## 1.2 Balancing memory and convergence rate

Fundamental tradeoffs exist between memory capacity, memory bandwidth, communication cost, and algorithmic characteristics of stability and convergence rate. The tradeoffs are generally resolved to minimize time to solution. GPUs attain high memory bandwidth advantages over CPUs at the cost of smaller memory capacity. Anderson extrapolation promotes fewer, more expensive steps, reusing cached state-vector data. In distributed memory implementations, it produces convergence with fewer interprocessor communication steps. It has tuning parameters such as window size and damping that can be tuned to application and architecture. We are assessing its utility in machine learning more broadly at a time of emergent CPU-GPU superchips.

## 1.3 Deep equilibrium neural network models

Deep equilibrium models (DEQs) are the continuum limit of explicit neural networks as the number of layers approaches infinity [26], approximating many explicit layers with a single, implicit layer with exponentially fewer parameters using a backward pass including the output. This reduces the inverse problem in parameter space to a fixed point iteration problem, enabling the usage of nonlinear, vector-to-vector mapping techniques to compute the fixed point iterations that converge to the deep equilibrium state parameters by minimizing the loss function. With gains in memory and acceleration, DEQs are fit for large-scale computer vision and natural language processing tasks and benefit more from matrix-vector operation-optimized computing architectures like GPUs and CPU-GPU superchips.

The standard approach using forward iteration for fixed point iteration problems often does not efficiently converge to the fixed point and suffers from initially slow error reduction and local minimum trapping in nonlinear problems like deep neural networks. Anderson extrapolation outperforms

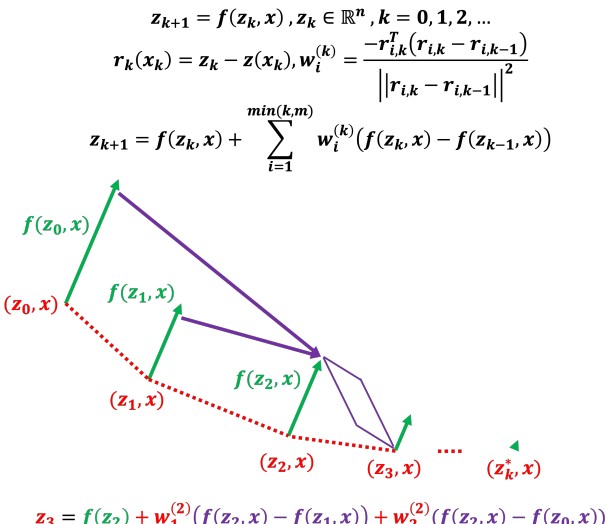

$$z_{k+1} = f(z_k, x), z_k \in \mathbb{R}^n, k = 0, 1, 2, \ldots$$

$$r_k(x_k) = z_k - z(x_k), w_i^{(k)} = \frac{-r_{i,k}^T(r_{i,k} - r_{i,k-1})}{\left\|r_{i,k} - r_{i,k-1}\right\|^2}$$

$$z_{k+1} = f(z_k, x) + \sum_{i=1}^{min(k,m)} w_i^{(k)}(f(z_k, x) - f(z_{k-1}, x))$$

$$z_3 = f(z_2) + w_1^{(2)}(f(z_2, x) - f(z_1, x)) + w_2^{(2)}(f(z_2, x) - f(z_0, x))$$

Figure 3: Mathematical formulation and vector representation. Adapted from Y. He & H. De Sterck. "Linear Asymptotic Convergence Analysis of Anderson Acceleration, with Krylov Formulation in the Linear Case" Copper Mountain Conference (2022), ICERM Workshop (2023). Available at: `https://www.bilibili.com/video/BV1Wa411i77y/` and `https://icerm.brown.edu/video_archive/?play=3320`

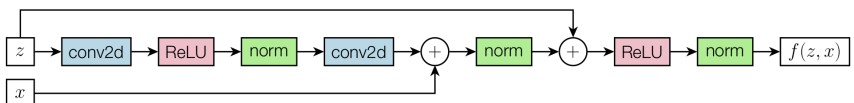

Figure 4: Deep equilibrium neural network model architecture (Source: NeurIPS Tutorial, 2020 [35]). $f(z, x) = \mathrm{norm}(\mathrm{ReLU}(z + \mathrm{norm}(x + W_2 * (\mathrm{norm}(\mathrm{ReLU}(W_1 * z))))))$. "norm" here is a group norm, representing a statistical normalization [32].

standard forward iteration by combining information from previous iterations to span a searchable subspace to extrapolate the next iteration, enhancing convergence rates at the expense of memory usage in each iteration. When the original fixed point iteration is contractive and thus guaranteed to converge, Anderson is theoretically guaranteed not to be slower [30] and it experimentally observed to be considerably faster in numerous applications.

DEQs represent any neural network at arbitrary depths and connectivities with a single implicit layer consuming vastly fewer parameters with faster forward passes for accelerated training and inferences. The implicit function theorem shows how gradients can be computed in the DEQ framework, facilitating backpropagation through the equilibrium state [9, 35].

DEQs provide a framework for accelerating deep learning, extending the capacity of deep networks within a single-layer architecture through fixed point computations and advanced root-finding algorithms. Their amenability to convergence acceleration with techniques like Anderson positions DEQs as a robust method to reduce computation needed to build state-of-the-art models and scale up beyond current computational limitations.

## 2 Methods

This work demonstrates Anderson extrapolation to accelerate AI performance algorithmically without increasing processors. Since it does not require inverting matrices approximating Hessians of the dimension of the state space, but only matrices of the dimension of the Anderson window size, it benefits from hardware optimized for uniform vector operations, like GPUs. We benchmark Anderson acceleration against standard forward iteration on GPUs and CPUs.

### 2.1 Mathematical formulation

Fixed point acceleration starts with the fixed point iteration formula $z^\star = f(z^\star, x)$. Forward iteration, $z^{k+1} = f(z^k, x)$, moves step-wise towards this fixed point.

**Algorithm 1** Extrapolation for Fixed Point Iteration [35]

---
**Input:** Function $f$, initial guess $x_0$, window size $m = 5$, regularization $\lambda = 1e - 5$, max iterations
$max\_iter = 1000$, tolerance $tol = 1e - 2$, mixing parameter $\beta = 1.0$
Initialize $n$, batch size $b$, channels $d$, height $H$, width $W$ from $x_0.shape$
$X, F \leftarrow$ Initialize tensors based on $b$, $m$, and $d \times H \times W$
$H, y \leftarrow$ Initialize for least squares solver
$times, res \leftarrow$ Initialize lists for timing and residuals
**for** $k = 2$ **to** $max\_iter$ **do**
   Start timing this iteration
   $n \leftarrow \min(k, m)$
   $G \leftarrow F[:, : n] - X[:, : n]$
   Update $H$ using $G$
   Solve linear system for $\alpha$
   Update $X$ and $F$ using $\alpha$, $m$, and $\beta$
   Compute residual, $\frac{\|f(z^k, x) - z^k\|_2}{\|f(z^k, x)\|_2 + \lambda}$
   Store time and residual
   Check for convergence
   **if** residual $<$ tol **then**
     **break**
   **end if**
**end for**
**return** $X[:, k\%m](x_0)$, residuals, times

---

Anderson acceleration uses a linear combination of prior iterates, $z^{k+1} = \sum_{i=1}^{m} \alpha_i f(z^{k-i+1}, x)$, optimizing $\alpha_i$ to minimize the residual norm, $\frac{\|f(z^k, x) - z^k\|_2}{\|f(z^k, x)\|_2 + \lambda}$, leading to faster convergence. The coefficients must sum to unity, thus:

$$\text{minimize}_\alpha \quad \|G\alpha\|_2^2, \quad \text{subject to} \quad 1^T\alpha = 1 \tag{1}$$

The matrix $G$ is defined as:

$$G = \left[ f(z^k, x) - z^k, \cdots, f(z^{k-m+1}, x) - z^{k-m+1} \right] \tag{2}$$

The Lagrangian incorporating the equality constraint is:

$$L(\alpha, \nu) = \|G\alpha\|_2^2 - \nu(1^T\alpha - 1) \tag{3}$$

To solve for $\alpha_i$, we set up and solve:

$$\begin{bmatrix} 0 & 1^T \\ 1 & H \end{bmatrix} \vec{y} = \begin{bmatrix} 0 & 1^T \\ 1 & G^TG + \lambda I \end{bmatrix} \begin{bmatrix} \nu \\ \alpha \end{bmatrix} = \begin{bmatrix} 1 \\ 0 \end{bmatrix} \tag{4}$$

Anderson acceleration generally includes a mixing parameter $\beta$, incorporating some inertia when $\beta < 1$:

$$z^{k+1} = (1 - \beta) \sum_{i=1}^{m} \alpha_i z^{k-1+1} + \beta \sum_{i=1}^{m} \alpha_i f(z^{k-i+1}, x) \tag{5}$$

## 2.2 Dataset description, compute environment, and training details

The CIFAR10 dataset, with 60,000 32x32 labeled images in 10 classes, is used for supervised learning and image classification tasks. Accuracy is the ratio of correctly predicted labels to the total images, using cross-entropy loss.

High-dimensional tensors in standard PyTorch format are used. The compute environment includes Google Colab Pro with NVIDIA Tesla V100 GPUs and Intel Xeon CPUs. Training uses default hyperparameters from Kolter et al. [35] for comparison with prior results [7–10, 17, 24], with Anderson parameters $m = 5$ and $\beta = 1$.

## 2.3 Deep neural networks, deep equilibrium models, and fixed Point equations

Traditional neural networks use layer-wise transformations:

$$\begin{aligned} z_1 &= x \\ z_{i+1} &= \sigma(W_i z_i + b_i), \quad i = 1, \ldots, k - 1 \\ h(x) &= W_k z_k + b_k \end{aligned}$$

DEQs model a network as an infinitely deep system, finding a fixed point $z^\star$ that satisfies:

$$z^\star = \sigma(Wz^\star + Ux + b) \tag{6}$$

Here, $W$, $U$, and $b$ are shared across all layers, and $\sigma$ is the activation function. Solving for $z^\star$ avoids computing individual layers, reducing computational cost.

### 2.4 GPU Optimization and Parallelization

Anderson acceleration maps well to GPUs, suited for uniform tasks with high throughput.

Table 1: Summary of algorithmic improvements to training and inference without augmentation.

|  | Algorithm | DEQ (ours) | DEQ [Implicit] [9] | ResNet-18 [Explicit] [18] |
|---|---|---|---|---|
| Number of parameters | Standard | **64,842** | ~170,000 | ~170,000 |
|  | Accelerated | **64,842** | - | - |
| Training accuracy | Standard | 64.7% | - | - |
|  | Accelerated | **96.3%** | - | - |
| Testing accuracy | Standard | 64.2% | 82.2% | 81.6% |
|  | Accelerated | **79.1%** | - | - |
| Training time [seconds] | Standard | $1.2 \times 10^4$ | - | - |
|  | Accelerated | $\mathbf{1.4 \times 10^3}$ | - | - |
| Inference time [seconds] | Standard | 1 | - | - |
|  | Accelerated | **0.5** | - | - |
| Speedup relative to standard | Ratio | **2-8.6** | - | - |
|  | Compute saved | **50-88%** | - | - |

## 3 Results

Anderson extrapolation has a higher cost per iteration, measured in function evaluations or epochs. The main benefit is that Anderson extrapolation exhibits less fluctuation in accuracy, as seen in the test accuracy, whereas forward iteration shows more significant ups and downs in both training and testing accuracy, potentially indicating overfitting. Anderson acceleration reaches a higher accuracy plateau for both training and test datasets, suggesting better generalization capability. We speculate that this is due to better avoidance of suboptimal local minima because of the wider window of trial vectors from which each step is drawn.

Anderson extrapolation is benchmarked against traditional forward iteration methods in DEQs to understand its role in AI and HPC. The computational demand of Anderson extrapolation correlates with the number of epochs, as shown in Fig. 5. A trade-off is shown between accuracy and computing time, whereas forward iteration maintains a more consistent computational time as the number of epochs increases.

Implicit neural network model architecture performance is analyzed with the goal of understanding how incorporating Anderson acceleration impacts model accuracy and performance. The stability of train and test accuracy is observed, and Anderson acceleration demonstrates higher consistency over numerous epochs, whereas forward iteration reveals significant swings in train and test accuracy. Initialization error with Anderson is lower than with forward iteration.

We observe Anderson acceleration reaches higher accuracies in training and testing in less time than forward iteration. Anderson acceleration is also superior with random inputs. Across testing with random inputs, Anderson acceleration consistently outperforms or matches forward iteration, depending on target relative residual accuracy.

## 4 Discussion

These results show that Anderson extrapolation can train DEQ networks to higher accuracy than forward iterations and reach a given high accuracy in less time. Anderson extrapolation is also efficiently implementable in GPU programming environments, utilizing memory austerity and operational uniformity attributes similar to the forward algorithm. For large-scale neural network training problems requiring distributed memory, this study motivates porting and testing on state-of-the-art GPU architectures, CPU-GPU superchips, and emerging computing hardware.

GPUs have been shown to accelerate Anderson extrapolation beyond what could be achieved with standard forward iterations or with Anderson on CPUs. This is notable before reaching the 'crossover point,' the trade-off between computation speed and accuracy, illustrated in Figs. 1 and 6.

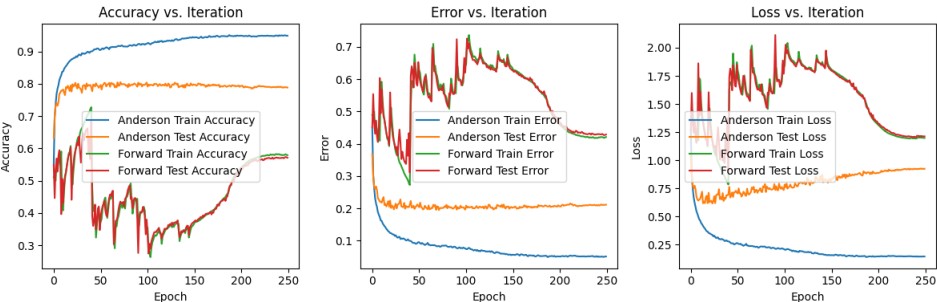

Figure 5: Evaluating CIFAR10 dataset through deep equilibrium. Anderson is 1.2x more accurate at stable convergence above mixing penalty.

The 'mixing penalty' due to the additional computational cost associated with Anderson acceleration is offset by the parallel processing capabilities of GPUs, enabling faster convergence than with CPUs or standard forward iterations alone.

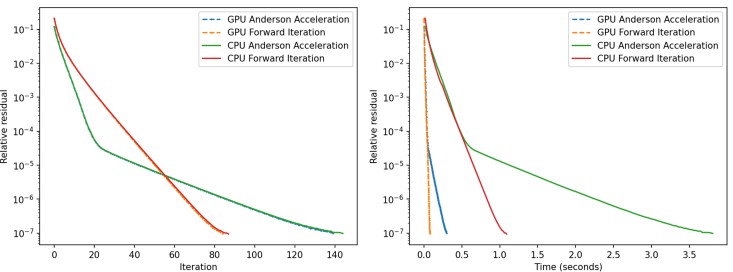

Figure 6: Evaluating relative residual, $\frac{\|f(z^k,x)-z^k\|_2}{\|f(z^k,x)\|_2+\lambda}$, for a random input $x$. A typical GPU is approximately 100-150x faster to target relative residual than a typical CPU using Anderson, with a mixing penalty that is approximately $10^{-1}$ to $10^{-2}$ lower.

The increase in time per iteration with Anderson arises from the residual minimization process during each acceleration step. The higher plateau for accuracy with Anderson compared to forward iteration suggests more robust learning when taking previous iterations into account. Monitoring the slowing of Anderson acceleration and switching to approximate forms of Newton's method (e.g., quasi-Newton, modified Newton, or inexact Newton) can be beneficial.

The unstable behavior with forward iteration necessitates lower learning rates and more epochs for training, increasing the time needed to reach the same accuracies achieved with Anderson by up to an order of magnitude. The inconsistency in accuracy with forward iteration raises concerns about overfitting during training, undermining the model's ability to generalize for reliable predictions on new, unseen data.

These findings indicate that Anderson acceleration improves DEQ performance with more rapid error reduction at the outset, as shown in Fig. 6 and Fig. 7. The rate at which peak accuracy is reached with extrapolation enables peak neural network performance in a fraction of the time required for forward iterations to stabilize at comparable accuracy. This acceleration is beneficial in time-sensitive applications where rapid deployment of accurate AI models is essential.

## 5   Conclusion

The integration of Anderson acceleration within deep learning workflows presents substantial improvements in computational efficiency, accuracy, and generalizability of implicit neural networks. Porting and parallelizing matrix-free acceleration techniques onto emerging CPU-GPU hybrid architectures holds promise. The accuracy and speed of deep equilibrium neural network training and inferences could be improved further, making them more viable for real-world applications beyond the classification task demonstrated herein. Based on investigations of explicit and implicit memory requirements [26], optimizations based on an Anderson-accelerated, fixed-point iteration implicit memory approach [35] are effective in memory-intensive computer vision processing, reducing memory and bandwidth consumption without compromising performance [26].

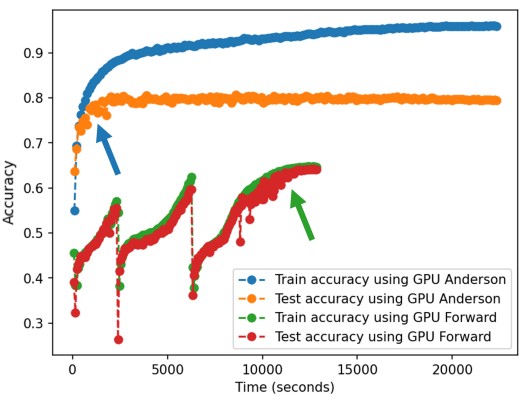

Figure 7: Deep equilibrium model is approximately 10x faster to stable convergence with Anderson relative to standard forward iteration, per Table 1

These methods applied to implicit neural networks, particularly DEQs, reveal new directions for AI research, such as exploring further acceleration gains from stochastic variants of Anderson extrapolation [31]. Exploiting the continuum limit of infinite explicit layers in implicit networks reduces memory usage and achieves favorable performance trade-offs [9], where gradient approximations, such as truncated backward gradient for backpropagation [16, 24], can be applied for even more acceleration.

## 6 NeurIPS Limitation and Broader Impact Statements

These results do not comprehensively search the Anderson hyperparameter space, nor do they establish the multiprocessor scalability at which they are aimed. Saving training and inference time and energy is the broader impact envisioned for this work. Being algorithmic in nature, it has the same potential for applied use and misuse as neural networks in general.

## Acknowledgments and Disclosure of Funding

SAA acknowledges funding from the KAUST Fellowship and the Extreme Computing Research Center throughout the duration of this work. SAA wishes to express sincere gratitude to Henning Soller, Mohamed-Slim Alouni, Matteo Parsani, George Turkiyyah, and Frédéric Laquai for their invaluable chats throughout this research. Special gratitude is extended to my family—late grandparents, parents, wife, daughter, and brothers—and to my friends for their unwavering support and encouragement throughout the duration of this study.

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
