# OpenReview forum: "Accelerating AI Performance using Anderson Extrapolation on GPUs"
_NeurIPS.cc/2024/Workshop/MLNCP — MLNCP Poster_

### Official Review · Reviewer_njWZ · 2024-09-23
**Anderson Extrapolation for Accelerating AI applications in Hardware**

**Rating:** 7
**Confidence:** 2

**Review:**

The paper propose to apply Anderson extrapolation, a nonlinear fixed point iteration technique to accelerate the computation workflow of  implicit neural networks (i.e. Deep Equilibrium Models). The results shows significant speed up on implicit neural networks with better accuracy which indicates the effectiveness of proposed method.

Strong point:
- Accelerating AI applications is an important topic, and this paper presents an promising direction.
- Many deep learning acceleration techniques has a implicit trade-off between efficiency and accuracy (i.e. quantization, pruning), however, this method not only addresses the efficiency issue but also improves accuracy.
- Evaluation although limited but is on point and concise in demonstrating the advantages of proposed method.

Weak points:
- The paper is generally well structured but very dense and somewhat difficult to read.
- I hope the authors would consider to provide some intuition for the gain in accuracy. It is logical that Anderson extrapolation leads to faster convergence hence better efficiency, but where does the gain in accuracy come from.

---

### Decision · Program_Chairs · 2024-10-10

Accept (Poster)